# Heterologous immunization with inactivated vaccine followed by mRNA-booster elicits strong immunity against SARS-CoV-2 Omicron variant

Fanglei Zuo [1,14], Hassan Abolhassani [1,14], Likun Du[1,14], Antonio Piralla [2,14], Federico Bertoglio [3], Leire de Campos-Mata[1], Hui Wan [1], Maren Schubert[3], Irene Cassaniti [2], Yating Wang[1], Josè Camilla Sammartino [2], Rui Sun[1], Stelios Vlachiotis[1], Federica Bergami[2], Makiko Kumagai-Braesch[4], Juni Andréll[5], Zhaoxia Zhang[6], Yintong Xue[7], Esther Veronika Wenzel [3,8], Luigi Calzolai[9], Luca Varani [10], Nima Rezaei[11], Zahra Chavoshzadeh[12], Fausto Baldanti[2,13,15], Michael Hust [3,15], Lennart Hammarström[1,15], Harold Marcotte[1,15] & Qiang Pan-Hammarström [1,15 ✉]

The recent emergence of the Omicron variant has raised concerns on vaccine efficacy and the urgent need to study more efficient vaccination strategies. Here we observed that an mRNA vaccine booster in individuals vaccinated with two doses of inactivated vaccine significantly increased the plasma level of specific antibodies that bind to the receptor-binding domain (RBD) or the spike (S) ectodomain (S1 + S2) of both the G614 and the Omicron variants, compared to two doses of homologous inactivated vaccine. The level of RBD- and S-specific IgG antibodies and virus neutralization titers against variants of concern in the heterologous vaccination group were similar to that in individuals receiving three doses of homologous mRNA-vaccine or a boost of mRNA vaccine after infection, but markedly higher than that in individuals receiving three doses of a homologous inactivated vaccine. This heterologous vaccination regime furthermore significantly enhanced the RBD-specific memory B cell response and S1-specific T cell response, compared to two or three doses of homologous inactivated vaccine. Our study demonstrates that mRNA vaccine booster in individuals vaccinated with inactivated vaccines can be highly beneficial, as it markedly increases the humoral and cellular immune responses against the virus, including the Omicron variant.

[1] Department of Biosciences and Nutrition, Karolinska Institutet, Huddinge, Sweden. [2] Molecular Virology Unit, Microbiology and Virology Department, Fondazione IRCCS Policlinico San Matteo, Pavia, Italy. [3] Department of Biotechnology, Institute of Biochemistry, Biotechnology and Bioinformatics, Technische Universität Braunschweig, Braunschweig, Germany. [4] Division of Transplantation Surgery, CLINTEC, Karolinska Institutet at Karolinska University Hospital, Stockholm, Sweden. [5] Science for Life Laboratory, Department of Biochemistry and Biophysics, Stockholm University, Stockholm, Sweden. [6] Department of Aging Neurology orthopedics, Karolinska University Hospital Huddinge, Stockholm, Sweden. [7] Department of Immunology, Peking University Health Science Center, Beijing, China. [8] Abcalis GmbH, Science Campus Braunschweig-Süd, Inhoffenstr. 7, 38124 Braunschweig, Germany. [9] European Commission, Joint Research Centre, Ispra, Italy. [10] Institute for Research in Biomedicine, Università della Svizzera italiana (USI), Bellinzona, Switzerland. [11] Research Center for Immunodeficiencies, Pediatrics Center of Excellence, Children's Medical Center, Tehran University of Medical Sciences, Tehran, Iran. [12] Pediatric Infections Research Center, Mofid Children's Hospital, Shahid Beheshti University of Medical Sciences, Tehran, Iran. [13] Department of Clinical, Surgical, Diagnostic and Paediatric Sciences, University of Pavia, Pavia, Italy. [14] These authors contributed equally: Fanglei Zuo, Hassan Abolhassani, Likun Du, Antonio Piralla. [15] These authors jointly supervised this work: Fausto Baldanti, Michael Hust, Lennart Hammarström, Harold Marcotte, Qiang Pan-Hammarström. ✉email: qiang.pan-hammarstrom@ki.se

In the current stage of the pandemic, vaccination against severe acute respiratory syndrome coronavirus 2 (SARS-CoV-2) is one of the main strategies to protect against coronavirus disease 2019 (COVID-19) and promote a return to normality. There has been an unprecedented worldwide effort to develop safe and effective vaccines against SARS-CoV-2, which has resulted in the authorization of up to 30 vaccines based on different technologies and with different efficacy rates, 10 of which are approved by the World Health Organization (WHO, as of January 2022) (https://covid19.trackvaccines.org/agency/who/). Vaccines against SARS-CoV-2 have been engineered employing the main vaccine technologies currently available, including whole virus (inactivated), protein subunits, viral vectors, and nucleic acid strategies (mRNA and DNA)[1,2] (https://www.who.int/news-room/feature-stories/detail/the-sinopharm-covid-19-vaccine-what-you-need-to-know) (https://www.who.int/news-room/feature-stories/detail/the-sinovac-covid-19-vaccine-what-you-need-to-know) (https://www.who.int/emergencies/diseases/novel-coronavirus-2019). More than 11.6 billion vaccine doses have been administered worldwide, where inactivated vaccines (CoronaVac (Sinovac) and BBIBP-CorV (Sinopharm)) have the highest number of delivered doses (45% worldwide, 65–85% efficacy)[3,4]. Due to concerns of the waning of antibody responses after vaccination and the emergence of variants of concern (VOC)[5–7], more than 1.8 billion additional/boosting doses have been administered worldwide.

There is also a growing interest in the efficacy of heterologous vaccination strategies (https://ourworldindata.org/coronavirus), which could mitigate the effects of putative shortages of supply, change in recommendations regarding usage of specific vaccines, and migration of individuals between countries with different COVID-19 vaccine regimes. Moreover, increasing evidence supports the notion that heterologous vaccination strategies like inactivated vaccines followed by a vector-based or an mRNA vaccine, or a viral vector-based vaccine followed by an mRNA vaccine may provide good tolerability and an enhanced immune response, as compared to the homologous vaccine regimen[8,9] (https://www.who.int/publications/i/item/WHO-2019-nCoV-vaccines-SAGE-recommendation-heterologous-schedules), thus offering a better protection. Very limited knowledge, however, is available on the immunogenicity and efficacy of heterologous vaccination approaches involving the inactivated vaccines.

Apart from the inactivated vaccines, virtually all authorized vaccines have been designed to recognize the spike (S) glycoprotein of the Wuhan strain of SARS-CoV-2, since antibodies directed against the S protein confer potent neutralizing activity. Hence, VOC with mutations in their S protein may alter the effectiveness of the currently available vaccines[10]. Recent reports suggest that the efficacy of the most widely used immunization against different VOC may be markedly reduced[10,11]. The recent emergence of the Omicron variant in South Africa in November 2021 and its rapid spread worldwide has strengthened concerns on vaccine efficacy due to its large number of mutations in the S protein, including 15 in the receptor-binding domain (RBD) according to the published original sequence[12,13]. Although the disease associated with Omicron seems to be less severe[14–16], there is an urgent need to study more efficient vaccination strategies due to the high transmutability and the high rate of immune escape of this VOC[17].

Here, we investigated whether antibodies induced by vaccination with an inactivated vaccine, an mRNA vaccine or a combination of both inactivated and mRNA vaccines targeted the RBD of G614 SARS-CoV-2 strain as well as Beta, Delta, and Omicron VOC. In the same cohort, we also studied neutralization activity against these VOC and SARS-CoV-2-specific memory B and T cell responses in selected individuals. We found that two doses of the inactivated vaccine (BBIBP-CorV or CoronaVac), followed by a third dose of an mRNA vaccine (BNT162b2 or mRNA-1273), markedly increased the humoral and cellular immune responses to the SARS-CoV-2 G614 strain and potentially to all major VOC, including the currently circulating Delta and Omicron variants.

## Results

Anti-RBD IgG antibodies were measured by ELISA in 238 samples from 175 healthy volunteers, grouped based on vaccination history (Methods, Supplementary Table 1, Supplementary Fig.1). Since there were no significant differences in the immune responses elicited by BNT162b2 and mRNA-1273 vaccines (Supplementary Fig. 2A), or BBIBP-CorV and CoronoVac vaccines (Supplementary Fig. 2B), we merged the samples into mRNA or inactivated vaccine groups, respectively. We noted that antibody responses declined over time, and a significant reduction was observed about 3 months (85 days) post the second dose of the mRNA vaccines ($p \leq 0.0001$; Supplementary Figs. 2 and 3). Thus, all studied samples were further assigned to subgroups based on sampling time: early or late sampling time, i.e., either less or more than 85 days after the given dose. The specific IgG plasma antibody responses against G614-RBD in individuals vaccinated with two doses of the inactivated vaccine and subsequently boosted with one dose of an mRNA vaccine were significantly higher compared to individuals vaccinated with two doses of the homologous inactivated vaccine at both sampling times (6.3- and 17.2-fold for <85 or >85 days group respectively, $p \leq 0.0001$; Fig. 1). Similar trends were recorded in an analysis of subset of individuals where longitudinal samples were available (Supplementary Fig. 4). Of note, the level of specific antibodies in the heterologous inactivated/mRNA prime-boost vaccination group was similar to that observed in individuals receiving a homologous third dose of an mRNA vaccine, which is currently the most powerful immunization schedule to various variants including the Omicron VOC[18–21], or a booster mRNA vaccine after natural infection (Fig. 1). In comparison, a third dose of the inactivated vaccine did not further increase the anti-RBD IgG antibody response compared to two doses of inactivated vaccine, when the sampling times were matched (Fig. 1).

To evaluate whether individuals who received heterologous vaccination would mount an increased response against the circulating variants, including the newly emerged Omicron VOC, we tested the cross-binding activity of plasma IgG antibodies against the RBD of Beta, Delta and Omicron VOC. Similar to our previous observation on plasma from convalescent donors[22], the cross-binding activity was more pronounced against the Delta-RBD compared to the Beta- and Omicron-RBD in all the studied vaccination regimes. Furthermore, heterologous vaccination gave rise to a markedly increased cross-binding activity against Beta (9.1- and 26.7-fold), Delta (7.4- and 17.8-fold) and Omicron (9.9- and 18.6-fold) VOC compared to those who received only 2 doses of the inactivated vaccine both at early and late times after vaccination. Importantly, again, it reached a level similar to that detected in donors receiving a homologous third dose of mRNA vaccines or a booster mRNA vaccine after natural infection (Fig. 1 and Supplementary Fig. 5). As observed for the anti-RBD IgG response, one dose of an mRNA vaccine significantly increased the anti-S (ectodomain, S1 + S2) IgG titers in individuals vaccinated with two doses of the homologous inactivated vaccines (<85 or >85 days) for both G614 (10.5- and 100.3-fold) and Omicron (13.9- and 103.8-fold) (Supplementary Fig. 6). Independent of vaccination regimens, the decrease in cross-binding IgG antibody binding activity was lower against the Omicron S protein (1.5- to 2.8-fold) than against RBD (3.9- to 7.3-fold) (Supplementary Figs. 5 and 7).

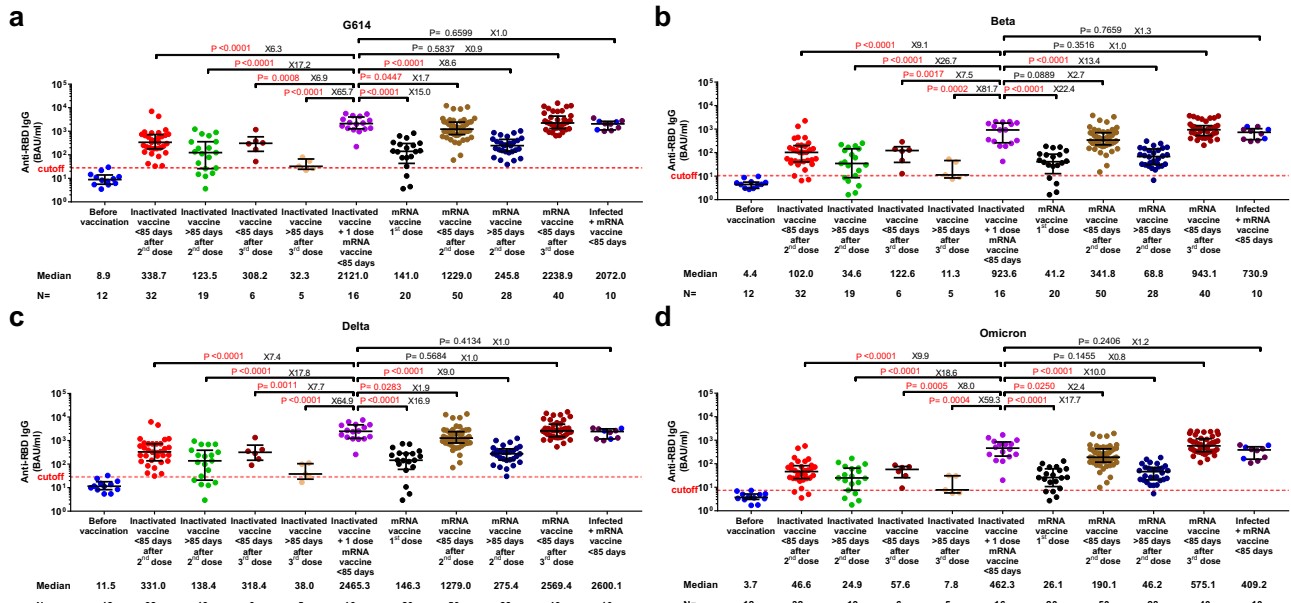

**Fig. 1 Level of specific IgG antibodies against receptor-binding domain (RBD) in different vaccination groups.** Comparison of anti-RBD against G614 SARS-CoV-2 (**a**) and variants of concern (**b–d**) presented as binding antibody units (BAU)/ml. Symbols represent individual subjects and horizontal black lines indicate the median. The cutoff-value (dashed red line) and number of fold differences of median between groups are indicated. For each group, the number of samples (N = ) and median antibody titers are shown below the X-axis. In the last group, convalescent donors (prior history of infection) were color-coded based on receiving one (purple) or two doses (blue) of mRNA vaccines. Whiskers indicate the interquartile range. Two-sided Mann–Whitney U test was used and P < 0.05 was considered statistically significant and marked with red color.

We furthermore measured the neutralizing antibody titers against the G614 virus and VOC in a subset of samples (n = 92) from the different vaccination groups (Fig. 2). Of note, the donors selected for this experiment were representative of the main study cohort with no significant differences in gender, age or specific antibody titers (Supplementary Table 2). In plasma specimens obtained from heterologous vaccinated individuals, the 90% neutralization titer (NT90) value against the G614 virus was significantly elevated compared to that after a second dose of inactivated vaccine (<85 days; median titer 640 vs. 100, 6.4-fold higher), and at a similar level as after a third dose of an mRNA (<85 days; median titer 640) or inactivated (median titer 640) vaccine. Furthermore, although at a reduced level compared to the G614 virus, plasma samples obtained from the heterologous vaccinated individuals showed enhanced ability to neutralize the Beta, Delta and Omicron VOC (median NT90 titers 80, 320 and 20 respectively), compared to those receiving two or three doses of inactivated vaccines (Fig. 2).

In addition, we measured the number of SARS-CoV-2-specific memory B and T cells by analyzing peripheral blood mononuclear cell (PBMC) samples available from a subset of the study subjects (n = 108) using ELISpot and FluoroSpot assays, respectively. The maximum value observed in the negative controls (non-infected individuals prior to vaccination) was set as a cutoff. We observed that the number of RBD-specific, IgG producing B cells was significantly higher in the heterologous vaccination groups than that in individuals with two-dose of homologous inactivated vaccine (30.8- and >73-fold higher respectively, in <85 and >85 days groups) or mRNA vaccine (2.9- and 5.5-fold higher respectively, in <85 and >85 days groups) (Fig. 3a). Moreover, the numbers of S1-specific, interleukin-2 (IL-2) and/or interferon-gamma (IFN-γ) producing T cells were increased to various levels (7.2- to 24.0-fold higher) in individuals vaccinated with the heterologous vaccine combination as compared to the two-dose homologous inactivated or mRNA vaccine (Fig. 3b–d). A similar pattern was observed for the SNMO peptide pool-specific T cells

derived from the spike protein (S), nucleoprotein (N), membrane protein (M), and the open reading frame (O) proteins (Supplementary Fig. 8). The positive impact of heterologous vaccination on specific memory B and T cells was significantly higher than in individuals who had been given three doses of the inactivated vaccine. Interestingly, while the number of RBD-specific IgG producing B cells in the heterologous vaccination group was similar to that in individuals with three doses of mRNA vaccine, or with mRNA vaccine booster after natural infection, the S1-specific T cell response (represented by the number of IL-2 or IL-2/IFN-γ producing T cells) induced by the heterologous vaccination seems to be the strongest among all vaccination/immunization strategies tested (Fig. 3a–d).

## Discussion

Vaccination provides prophylaxis against a variety of infectious diseases owing to the induction of neutralizing antibodies and cellular immunity against the pathogen. Vaccines can either be given in the form of inactivated virus, subunit protein vaccines or, more recently, mRNA-based vaccines. The recent SARS-CoV-2 pandemic has led to the development of a multitude of vaccine candidates, most of which target the S protein, utilizing different vaccine strategies.

We have previously shown that although plasma IgG antibodies against the RBD of the S protein of SARS-CoV-2 are markedly reduced 6–12 months after infection, long-lived B and T cell memory responses persist for up to 15 months and may thus aid in protection from re-infection and/or severe diseases[22,23]. Here, we showed that the decline of the plasma antibody levels was more rapid in the vaccinated individuals compared to naturally infected patients, supporting and extending previous findings[24,25]. This raises the question on strategies for boosting the immune response in vaccine recipients, which is of particular concern in view of the limited induction of cross-neutralizing antibodies against the Omicron VOC[20,26–30].

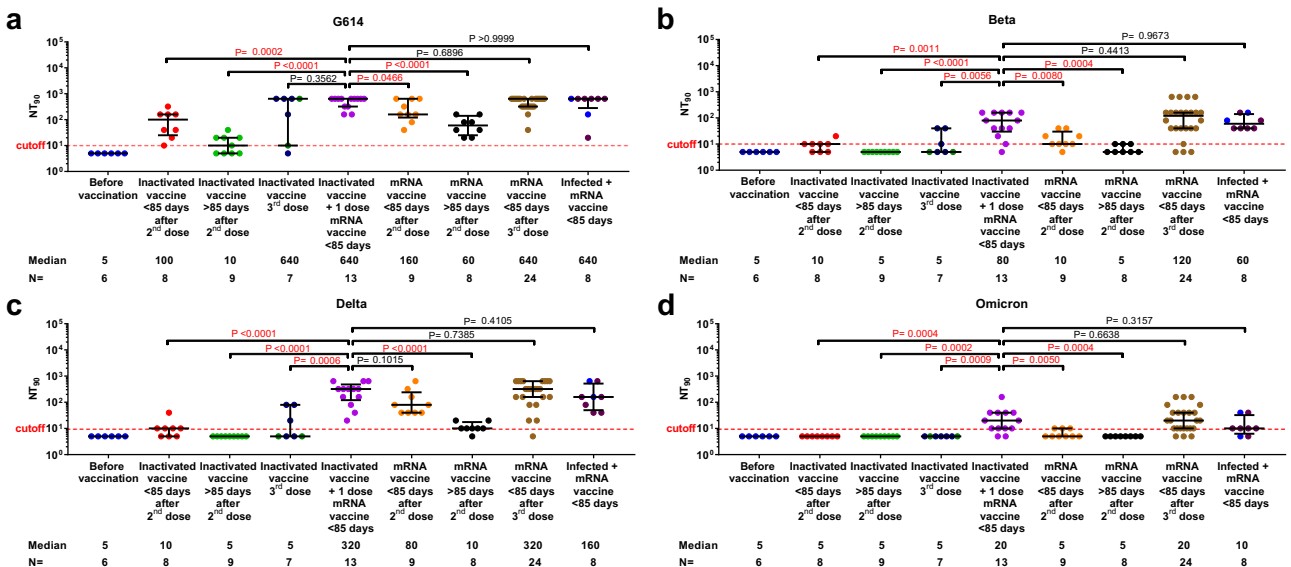

**Fig. 2 Plasma neutralization activity against SARS-CoV-2 in different vaccination groups.** Plasma neutralizing titers against Beta, Delta and Omicron variants of concern compared to the G614 virus (**a–d**). The figure shows the comparisons of 90% neutralizing titer (NT90) values in samples collected before vaccination and after different types of vaccinations. Symbols represent individual subjects and horizontal black lines indicate the median. For each group, the number of samples (N = ) and median antibody titers are shown below the X-axis. The dashed red line indicates the titer cutoff-value (≥1:10). The inactivated vaccine 3rd dose group was color-coded based on <85 days (dark blue) or >85 days (green). In the last group, convalescent donors (prior history of infection) were color-coded based on receiving one (purple) or two doses (blue) of mRNA vaccine. Whiskers indicate the interquartile range. Two-sided Mann–Whitney U test was used and P < 0.05 was considered statistically significant and marked with red color.

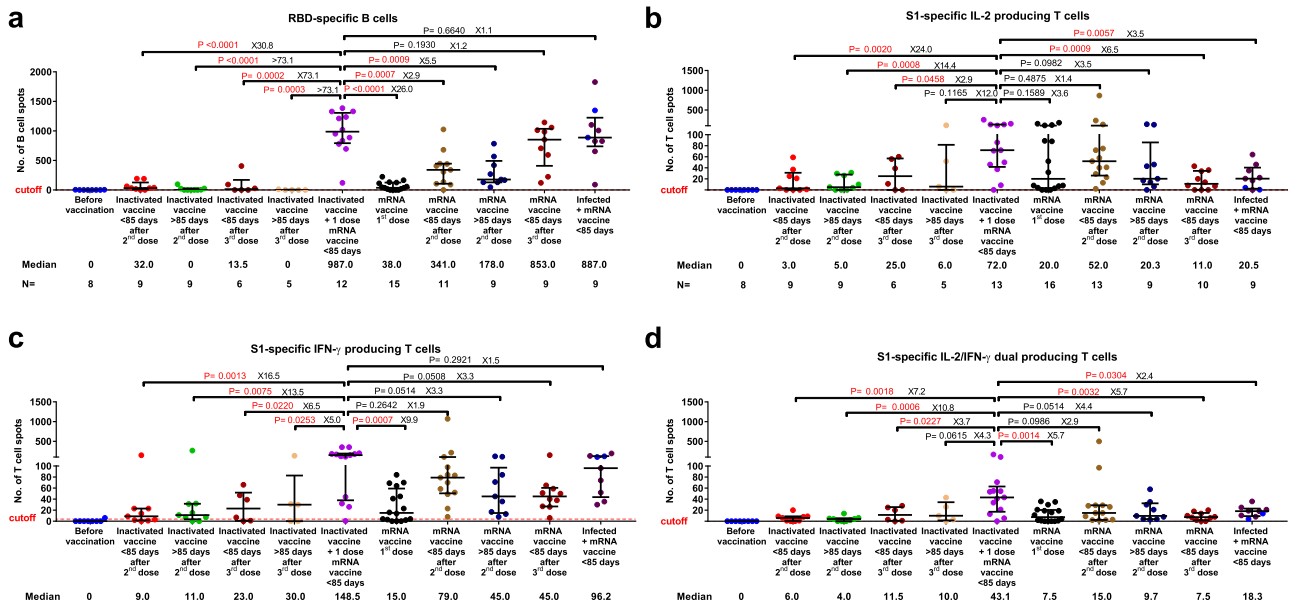

**Fig. 3 Specific memory B and T cell responses in different vaccination groups.** Receptor-binding domain (RBD)-specific, IgG producing memory B cell (**a**) and spike ectodomain 1 (S1)-specific T cell (**b–d**) responses in different groups of vaccinated individuals. Symbols represent individual subjects and horizontal black lines indicate the median. The cutoff-value (dashed red line) and number of fold differences of median between groups are indicated. As the median number of RBD-specific cells was 0 for two groups, the number of fold change differences compared to inactivated vaccine + one dose mRNA vaccine was estimated to be more than the highest calculated value (>73.1-fold). For each group, the number of samples (N = ) and median number of specific B cells or T cells are shown below the X-axis. In the last group, convalescent donors were color-coded based on receiving one (purple) or two doses (cyan) of mRNA vaccines. Whiskers indicate the interquartile range. Two-sided Mann–Whitney U test was used and P < 0.05 was considered statistically significant and marked with red color. IL-2: interleukin 2, IFN-γ: Interferon gamma.

Binding of the RBD of the S protein to the angiotensin-converting enzyme 2 (ACE2) receptor is a critical initial step for SARS-CoV-2 to enter into target cells. Most neutralizing antibodies that are produced following infection or vaccination target

the spike antigen on or proximal to the RBD and are considered important for blocking infection and viral clearance[31,32]. Furthermore, studies have found that anti-RBD IgG antibody and neutralization titers correlate across all variants following

infection or vaccination[22,33]. IgG against the RBD protein of the G614 strain of the virus in individuals who had received two doses of the inactivated vaccine showed only approximately one-third of the antibody levels compared to those obtained after two doses of an mRNA-based vaccine in both early and late sampling time points. This is in line with a previous head-to-head comparison of the two vaccines[34]. However, when the two groups were boosted with yet another dose of an mRNA-based vaccine, the specific antibody levels rose markedly in both groups and reached comparable levels, both being equal to the group of convalescent patients who had been given an mRNA-based vaccine boost. The head-to-head comparison study has also suggested that two doses of the inactivated vaccine induced a higher T cell response compared to the mRNA vaccine[34]. In our study, we observed that the number of RBD-specific memory B cells or S1-specific T cells in the blood of individuals receiving the heterologous vaccination is at par with (B cells), or even higher (T cells) than that in individuals with three doses of mRNA vaccine, or in convalescent patients boosted with an mRNA-based vaccine. Of utmost importance, we also observed that the heterologous vaccine-induced cross-binding and cross-neutralizing antibody titers against VOC as well as RBD-specific memory B cells or S1-specific T cells at a significantly higher level than three doses of inactivated vaccines.

Mutations in the RBD may lead to reduction in neutralization susceptibility of VOC by antibodies[35]. Therefore, anti-RBD antibody measurement can reflect the neutralization activity against emerging variants. The antibody levels against the Delta variant were equal to those against the G614 strain in all the groups tested (including those boosted by a third vaccine dose), whereas the antibody levels against the Beta, and in particular the Omicron variant, were much lower[21,36]. Importantly, the heterologous inactivated/mRNA prime-boost vaccination gave rise to a markedly increased cross-binding activity and neutralizing titer against all tested variants. The potent anti-RBD antibody and neutralizing response against the Omicron variant observed following booster vaccination may be a consequence of affinity maturation following initial vaccination or SARS-CoV-2 infection with the ancestral S protein, increasing the affinity of existing neutralizing antibodies to counteract the effect of mutations in their target epitopes, and/or may be due to stimulation of low-level neutralizing antibodies that target conserved epitopes on the S protein[20,37]. This may thus equalize the RBD-specific antibody response and neutralization activity against the highly mutated Omicron variant and the original parent strain explaining the higher increase of RBD-specific antibody response against Omicron (9.9-fold) compared to the original strain (6.3-fold) following boosting.

As previously reported, we observed a more pronounced loss of IgG antibodies against the Omicron RBD than the S protein which might be due to the higher density of mutations in the RBD[38]. The level of those S-specific antibodies increased following the heterologous inactivated/mRNA prime-boost vaccination and could correspond to both neutralizing or non-neutralizing antibodies mediating constant fragment (Fc) effector functions[38].

B cell reactivity against RBD of different VOC has not been analyzed in great detail. As the heterologous vaccination approach significantly boosts the overall level of specific memory B cell response against the G614 virus, it is likely that these B cells can be quickly recalled and produce the amounts of specific antibodies needed to combat the infection. In addition, B cells with lower affinity binding to VOC can continue to evolve through affinity maturation, thus giving rise to better protection[39–41].

Very recently, it has been suggested that the T cell response against the Omicron is less affected than the antibody

responses[39,42,43]. Nevertheless, we showed here that the heterologous inactivated/mRNA prime-boost vaccination scheme gave a strong boost for the overall T cell response against the S protein. Our results suggest that this heterologous vaccination strategy may have advantageous in eliciting a broader immune response, although further investigations are required.

Individuals who received the inactivated vaccine and subsequently relocate to countries in which the inactivated vaccine is not generally used need to start the vaccination process with a vaccine approved in their host country. For the mRNA vaccination, the recommended interval between the first and second dose of mRNA vaccines is generally 4–6 weeks with a boost after 6 months. Preliminary data show that a second mRNA boost around 5 weeks after the first one did not further increase the immune response in individuals that received the inactivated vaccine and additional studies should be considered to determine when the second boost must be given.

As of February 2022, the Omicron variant, B.1.1.529 or BA.1, is constantly evolving and considered to have divided into four lineages: BA.1, BA.1.1, BA.2, and BA.3 with BA.1 accounting for most of the Omicron-cases worldwide but where BA.2 gaining ground and becoming the dominant form in some countries[44]. BA.1, BA.2, and BA.3 share 12 RBD mutations, and BA.2 and BA.1 have 3 and 4 additional RBD mutations, respectively[45]. A recent study showed that neutralization of BA.2 by plasma from individuals vaccinated and boosted with the mRNA BNT162b2 reached titers only slightly (1.4-fold) lower than that against BA.1, suggesting that heterologous vaccination with inactivated and mRNA vaccines may still be efficient against new emerging variants[46].

Taken together, our results suggest that a booster dose of an mRNA vaccine to individuals who have received two doses of the inactivated vaccines strongly augments the specific antibody levels, neutralization activity, and memory B and T cell recall responses against the SARS-CoV-2 virus and VOC including the new Omicron variant. The heterologous inactivated/mRNA prime-boosting regime may thus be a very promising vaccination strategy based on our immunogenicity study and the very recent suggestions by Perez-Then et al.[47] and Cheng et al.[48], and it is also supported by the recent real-world experience in Chile showing an increased effectiveness against COVID-19 and reduction of hospitalization using this strategy as compared to a homologous booster using the inactivated vaccine (https://cdn.who.int/media/docs/default-source/blue-print/chile_rafael-araos_who-vr-call_25oct2021.pdf?sfvrsn=7a7ca72a_7). The heterologous inactivated/mRNA prime-boosting strategy was also shown to be safe and cause fewer adverse reactions and events compared to other homologous and heterologous vaccination regimens[49]. Given that the inactivated vaccines contributed more than 45% of vaccine doses distributed worldwide and most of the individuals receiving these vaccines are living in developing countries, an improved strategy based on inactivated vaccines is likely to be highly beneficial for billions of people and thus our fight against both the present and potentially forthcoming VOC. It should be emphasized that with one mRNA vaccination as a boost (following two doses of an inactivated vaccine) similar virus-neutralizing titers against ancestral as well as VOC can be achieved as if three vaccinations using mRNA would be used. Thus, one additional mRNA dose is sufficient to come to the "gold-standard" response and would be a good investment even in resource-poor conditions to achieve the global population protection against this still very harmful disease.

Limitations of this study include a rather low number of total participants and limited access to prospective sample collection. The volunteers included in the study also tend to be younger than the average global population. Furthermore, the data from

various vaccination groups were mainly compared using cross-sectional analysis and longitudinal analysis was performed only in a subset of samples. Hence, our results should be confirmed in larger-scale longitudinal studies with different age groups and where the fluctuation of pre- and post-vaccination titers in specific individuals would be compared.

## Methods

**Study design**. Study inclusion criteria included subjects being above 18 years of age, received inactivated and/or mRNA vaccines and have vaccination history documented (type of vaccine, number of doses, interval between the doses, days after the latest dose, if have been infected), and who were willing and able to provide written informed consent. The study includes 238 samples from 175 healthy volunteers (58.8% females, median age of 36 years) in Sweden ($n = 101$), Germany ($n = 18$) Iran ($n = 34$) and Italy ($n = 22$) during 2021–2022. Individuals were followed at 1 ($n = 142$), 2 ($n = 22$), 3 ($n = 9$) or 4 time points ($n = 2$) during their respective vaccination schedule (Supplementary Fig. 1). The samples were further characterized based on the vaccination record: homologous inactivated vaccination (BBIBP-CorV, $n = 42$, 45 samples; CoronaVac, $n = 7$, 8 samples), homologous mRNA vaccination (BNT162b2, $n = 94$, 123 samples; mRNA-127, $n = 8$, 15 samples), heterologous vaccination with two doses of inactivated vaccine followed by an mRNA vaccine boost at 4–16 months ($n = 16$, 9 samples before and 16 samples after booster), and homologous mRNA vaccination preceded by a prior history of mild SARS-CoV-2 infection based on self-reported or laboratory evidence ($n = 8$, 10 samples). Plasma samples from pre-vaccinated, non-infected healthy donors from our cohort ($n = 12$, Supplementary Fig.1) were also collected as negative controls. The study was approved by the ethics committees in institutional review board of Stockholm, Technische Universität Braunschweig, the Tehran University of Medical Sciences, and the Policlinico San Matteo.

**Production of SARS-CoV-2 RBD and spike ectodomain protein**. The RBD and spike ectodomain (S1 + S2, referred as S) sequence of the Omicron variant was ordered as GeneString from GeneArt (Thermo Fisher) according to EPI_ISL_6590608 (partial RBD Sanger sequencing from Hong Kong), EPI_ISL_6640916, EPI_ISL_6640919 and EPI_ISL_6640917 including Q493K which was corrected later to Q493R. All sequences of the RBD (319-541 aa of GenBank: MN908947) and spike ectodomain (14-1208 aa of GenBank: MN908947 with proline substitutions at positions 986 and 987 and "GSAS" substitution at the furin site, residues 682–685) of G614 and Omicron BA.1 were inserted in a *NcoI/NotI* compatible variant of the OpiE2 expression vector containing an N-terminal signal peptide of the mouse Ig heavy chain and a C-terminal 6xHis-tag[50]. RBD of G614, Beta, Delta and Omicron and S of G614 and Omicron BA.1 were expressed baculovirus-free in High Five insect cells and purified on HisTrap excel columns (Cytiva) followed by preparative size exclusion chromatography on 16/600 Superdex 200 pg columns (Cytiva)[51,52].

**Detection of antibodies specific to SARS-CoV-2**. For assessing the anti-RBD IgG binding activity, high-binding Corning Half area plates (Corning #3690) were coated overnight at 4 °C with RBD derived from the G614, Beta, Delta and Omicron (1.7 µg/ml) variants in PBS. Serial dilutions of plasma in 0.1% BSA in PBS were added and plates were subsequently incubated for 1.5 h at room temperature. Plates were then washed and incubated for 1 h at room temperature with horseradish peroxidase-conjugated goat anti-human IgG (Invitrogen #A18805)(diluted 1:15 000 in 0.1% BSA-PBS). Bound antibodies were detected using tetramethylbenzidine substrate (Sigma #T0440). The color reaction was stopped with 0.5 M $H_2SO_4$ after 10 min incubation and the absorbance was measured at 450 nm in an ELISA plate reader. For each sample, the $EC_{50}$ values were calculated using GraphPad Prism 7.05 software and expressed as relative potency toward an internal calibrant for which the Binding Antibody Unit (BAU) was calculated using the WHO International Standard 20/136 in relation to the G614 RBD. The positive cutoff was calculated as 2 standard deviations (2 SD) above the mean of a pool of pre-vaccination samples ($n = 12$).

**Neutralization assay against authentic SARS-CoV-2**. SARS-CoV-2 G614 strain and VOC (Beta, Delta and Omicron) were isolated from patients in Pavia, Italy and identified through next-generation sequencing. A microneutralization assay was used to determine the titers of neutralizing antibodies against SARS-CoV-2 strains[53,54]. Briefly, 50 µl of plasma, starting from 1:10 in a serial twofold dilution series (up to 1:640), was added to two wells of a flat-bottom tissue-culture microtitre plate (COSTAR, Corning Incorporated), mixed with an equal volume of 100 median Tissue Culture Infectious Dose (TCID50) of a SARS-CoV-2 strain, previously titrated and incubated at 33 °C in 5% $CO_2$. All dilutions were made in Eagle's minimum essential medium with addition of 1% penicillin, streptomycin and glutamine and 5 µg/ml of trypsin. After 1 h of incubation at 33 °C in 5% $CO_2$, VERO E6 cells (VERO C1008 [Vero 76, cloneE6, Vero E6]; ATCC® CRL-1586¨) were added to each well. After 72 h of incubation at 33 °C in 5% $CO_2$, wells were scored to evaluate the degree of cytopathic effect compared with the virus control and stained with Gram's crystal violet solution (Merck) plus 5% formaldehyde 40%

m/v (Carlo ErbaSpA) for 30 min. Microtitre plates were then washed in running water. Crystal Violet staining indicates live cells and thus the presence of neutralizing antibodies. The neutralizing titer ($NT_{90}$) was the maximum dilution giving a reduction of 90% of the cytopathic effect. The cutoff for positivity was ≥1:10. Positive and negative controls were included in all test runs.

**ELISpot and FluoroSpot**. Peripheral blood mononuclear cells (PBMCs) were isolated from whole blood by standard density gradient centrifugation using Lymphoprep (Axis-Shield) following the manufacturer's instructions. PBMCs were then cryopreserved and stored in liquid nitrogen until analysis.

After thawing and washing, the cells were counted with trypan blue. PBMCs were incubated for 4 days in RPMI-1640 medium with 10% FCS, supplemented with the TLR7 and TLR8 agonist imidazoquinoline resiquimod (R848, 1 µg/ml; Mabtech AB, Nacka, Sweden), and recombinant human IL-2 (10 ng/ml) for stimulation of memory B cells. The ELISpot plates pre-coated with capturing monoclonal anti-human IgG antibodies were incubated with a total of 300,000 or 30,000 viable pre-stimulated cells per well for detection of RBD-specific IgG and total IgG (positive control) secreting cells, respectively. The number of B cells secreting SARS-CoV-2 RBD-specific IgG and total IgG were measured using the Human IgG SARS-CoV-2 RBD ELISpotPLUS kit (Mabtech AB)[23].

SARS-CoV-2 S1 and S N M O specific IFN-γ and/or IL-2-secreting T cells were detected using the Human IFN-γ/IL-2 SARS-CoV-2 FluoroSpotPLUS kit (Mabtech AB)[22,23]. The plates pre-coated with capturing monoclonal anti-IFN-γ and anti-IL-2 were incubated overnight in RPMI-1640 medium containing 10% FCS supplemented with a mixture containing a SARS-CoV-2 peptide pool (scanning or defined pools), anti-CD28 (100 ng/ml) and 300 000 viable cells per well in humidified incubators (5% $CO_2$, 37 °C). A polyclonal activator for human T cells (anti-human CD3 monoclonal antibody CD3-2, #3605-1, Mabtech) was used as a positive control for cytokine secretion. The SARS-CoV-2 S1 scanning pool contains 166 peptides from the human SARS-CoV-2 virus (#3629-1, Mabtech AB). The peptides are 15-mers overlapping with 11 amino acids, covering the S1 domain of the S protein (amino acid 13-685). The SARS-CoV-2 S N M O defined peptide pool contains 47 synthetic peptides binding to human HLA, derived from the S, N, M ORF3a and ORF7a proteins (#3622-1, Mabtech AB).

Results of ELISpot and FluoroSpot assays were evaluated using an IRIS-reader and analyzed by the IRIS software version 1.1.9 (Mabtech AB). The results were expressed as the number of spots per 300,000 seeded cells after subtracting the background spots of the negative control. The cutoff-value was set at the highest number of specific B- and T cell spots from the pre-vaccinated individuals. The number of SARS-CoV-2 specific T cells (per 300,000 cells) producing either IL-2, IFN-γ, or both IL-2 and IFN-γ (IL-2/IFN-γ were plotted).

**Quantification and statistical analysis**. Microsoft Excel 2017 was used for data collection for this study. Two-sided Mann–Whitney U test was used for comparisons between groups in anti-SARS-CoV-2 antibody levels and numbers of specific memory B and T cells. Two-sided Mann–Whitney U test, two-sided Fisher's exact test and Chi-square statistic were used for evaluation of demographic data and specific-IgG antibody responses in a subset of individuals selected for naturalization study compared to the main study cohort. All analyses and data plotting were performed using GraphPad version 7.05 or R version 3.6.1. A $p$ value < 0.05 was considered statistically significant.

**Reporting summary**. Further information on research design is available in the Nature Research Reporting Summary linked to this article.

## Data availability

All data supporting the findings of this study are available online in Zenodo at:// zenodo.org/record/6305550#.YhzpWmRKhaQ.

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

## Acknowledgements

This work was supported by The European Union's Horizon 2020 research and innovation program (ATAC, 101003650, L.C., L.V., F.BA., M.H., L.H., H.M., Q.P.H), the Center for Innovative Medicine at the Karolinska Institutet (Q.P.H), the Swedish Research Council (Q.P.H) and the Knut and Alice Wallenberg Foundation (KAW, L.H., Q.P.H).

## Author contributions

F.Z., H.A., L.H., H.M., and Q.P.H. conceived the study. L.D., H.A., A.P., F.BE., L.D.C., M.S., I.C., Y.W., J.C.S., R.S., S.V., F.BE, M.K., J.A., Z.Z., Y.X., E.V., L.C., L.V., N.R., Z.C., F.BA., M.H., and H.M. were involved in sample collection, processing and preparation for immunologic assays. F.Z., H.A., L.H were involved in interpretation of raw data. F.Z. and H.W. performed computational analysis and image preparation. L.H., H.M., and Q.P.H designed the combined laboratory protocols and supervised the entirety of the project. All authors approved the paper.

## Funding

## Competing interests

The authors declare no competing interests.

**Additional information**

