## [Peer Review File · Nature Communications]

Reviewer comments, initial review:

Reviewer #1 (Remarks to the Author):

There is a lot of interest in heterologous boost strategies, and if they are acceptable alternatives or maybe even better than homologous boosts. So the manuscript topic is important and timely. The manuscript by Zuo et al is stated to describe results from heterologous booster vaccines .

I really struggled to understand the study design. From what I can tell, these are 183 samples from 124 participants that received some mix of 4 different types of vaccines. The samples were collected either after the first vaccination, or some period of time after the second (unclear range, but exceeds 85 days), with some samples were after the third vaccination. Additionally, some healthy volunteer and convalescent samples were also assessed. With these samples, comparisons of the Elisa and neutralization assays were made among these various groups.

The biggest challenge is how to use these various data points to provide data to inform the scientific and public health communities. Showing post vaccine titers, without knowing pre-vaccine titers limits interpretation. I.e. how do we know that these groups are similar pre boost? There are some important questions in this field, but unfortunately the study as implemented is unable to contribute to a better understanding of heterologous boosts.

Specific comments:

1. Abstract, line 41- increased 6 fold and 14 fold compared to what?
2. Abstract, line 42-44. The two sentences seem in contradiction. The heterologous tiers were the same as homologous, but heterologous significantly increased B cell response. Please clarify. Please note what you are measuring, i.e. pseudovirus neutralization titers, rather than using colloquial phrases like "level of specific antibodies."
3. Introduction can be considerably abbreviated. This section should set up the reader by reminding them of key facts, but is not generally meant as an exhaustive review. For examples, the review of all the available vaccines, % of world supply, and efficacy, doesn't help the reader understand heterologous boosts. Same with review of all the worlds variants.
4. Line 110 – why did you use 85 days to characterize early vs late boost.
5. Line 220 – the design is not clear. There were 183 samples from 124 participants with vaccination schedule documented. What does that mean? The primary was documented? The boost was documented? or both?

Reviewer #2 (Remarks to the Author):

The submitted work describes heterologous vaccination strategies with the aim of improving response to the currently most active Omicron variant. This is quite important since this variant is exhibiting an immune escape phenotype and protection against infection is even reduced with the otherwise highly protective mRNA vaccines Comirnaty or SpikeVax. Since only a fraction of the global population has received those vaccines, strategies to increase the protection against Omicron are needed to improve world-wide vaccination supply.

The submitted manuscript describes such an approach, especially boosting of individuals that previously received two vaccinations with inactivated SARS-CoV-2 vaccines: BBIBP-CorV and CoronaVac. The administration of a single dose of Comirnaty did increase the vaccination response in those vaccinated individuals. Especially important is that a single vaccination of mRNA in those individuals reached specific antibody titer comparable to individuals that received three doses of mRNA vaccines. This indicates that such a third vaccination using mRNA might compensate for the lower immune response induced in those individuals previously receiving inactivated vaccine. However, at this stage it needs to be cautioned that three doses of the inactivated vaccines were not tested in this study and a comparison of vaccine potency of a such a third vaccination is not available.

I perceive this study as scientifically highly relevant for the ongoing SARS-CoV-2 pandemic and

ways to improve global protection against SARS-CoV-2 ancestral or VOCs by heterologous vaccination.

Moreover, for individual who got an inactivated vaccine, there is a need to know about the safety profile of a heterologous vaccination. As well this adds to the scientific knowledge.

The data generation and interpretation is sound.

Suggested Improvements:

Nevertheless, a weakness of this study is the absence of virus neutralization titers against ancestral and the most important variant Omicron, but for four data points displayed in supplemental figure 5. Please either generate more functional data in terms of VNTs or explain why this is not possible. Moreover, please indicate how those samples were selected and to which extent they reflect the overall study population? Are this high/mid/or low responding individuals?

The significance of RBD binding antibodies could be discussed more in details.

Minor comments:

To which extend is the study population reflecting the age distribution in the countries where the inactivated vaccines are used? The study population seems rather to be young for a european population. However, relevant is the age distribution in which the third vaccination using an mRNA vaccine would potentially be applied. Please comment.

line 51-52: ...vaccination... is the main strategy to protect against... : not sure if really true since at least early in the pandemic social distancy or "lockdowns" have been used. Either indate this is the main strategy since vaccines are available or make a disclaimer: "in the current stage of the pandemic"

line 86: comment on the mutations in spike RBD: the manuscript is describing Omicron as a variant with 15 mutations in spike, however, accurently variants with only 12 mutations (compared to ancestral spike) have been reported. So, already at this point Omicron is changing again.

line 87: the BA.2 lineage of Omicron might even have a high transmissibility, since (even if slowly) increasing in areas where Omicron is predominant.

line 118-119: "currently the most powerful immunization schedules to various variants 118 including Omicron VOC". Please provide a reference for this statement.

line 127: Any ideas why the response would be stronger boosted by a third vaccination against Omicron, since this is the most divergent variant? Please comment.

line 187: the neutralization data is available only for a very small subset of study participants. Please indicate this at this point of the manuscript.

line 206: please explain in few words about why the real-world data from Chile is supportive in the context of this manuscript. Please comment.

line 213: please indicate why the lack of data for a third vaccination using inactivated vaccine is not a limitation of the study? Please comment.

Reviewer #3 (Remarks to the Author):

In this work the authors have compared serum antibody ELISA titers against the receptor binding domain (RBD) from Wuhan-like SARS-CoV-2 with those against RBDs of VOCs, including omicron obtained from individuals vaccinated with inactivated COVID-19 vaccines, mRNA vaccines, and homologous or heterologous booster vaccination regimens (inactive + mRNA, mRNA + mRNA, convalescent + mRNA). Antibody ELISA titers against wild type and VOC RBD after mRNA booster was similar in individuals that were primed by inactivated virus vaccines, mRNA vaccines or in convalescent individuals. This correlated with enhanced levels of B cells and T cells as measured by ELISPOT.

Minor comment:

Line 77: The authors refer to Wuhan-like viruses that circulated early during the pandemic as "Wild type" virus. Ideally the authors should make clear in the beginning that wild type virus refers to Wuhan-like viruses that circulated before variants of concern started circulating.

Major comment:

In order to understand the relevance for protection of enhanced levels of antibody binding to RBD in ELISA assays, microneutralization assays with more sera (pseudo- or real viruses) need to be done. Also, antibodies that bind outside of the RBD can be protective. This should be measured as well, for example by ELISA against whole spike protein.

The authors have generated much data that is very interesting to the scientific community (for example the antibody decline over time and the microneutralization data, be it with limited groups size). The manuscript would benefit from it if these data were to be moved into the main text.

2 March 2022

Reviewer #1:

There is a lot of interest in heterologous boost strategies, and if they are acceptable alternatives or maybe even better than homologous boosts. So, the manuscript topic is important and timely. The manuscript by Zuo et al is stated to describe results from heterologous booster vaccines.

Reply: We appreciate the reviewer's comments.

I really struggled to understand the study design. From what I can tell, these are 183 samples from 124 participants that received some mix of 4 different types of vaccines. The samples were collected either after the first vaccination, or some period of time after the second (unclear range, but exceeds 85 days), with some samples were after the third vaccination. Additionally, some healthy volunteer and convalescent samples were also assessed. With these samples, comparisons of the Elisa and neutralization assays were made among these various groups.

Reply: The study was originally planned to follow up the antibody response, T- and B-cell response after 2 doses of vaccination (first time-point at 3-4 weeks after vaccination, then second time-point 3 months after the vaccination) and whenever is possible, additional samples were also collected from the same donors (before vaccination, or after 1 dose of mRNA vaccine). During the study, the vaccination schedule and type of vaccines were changed in the respective countries and some of our donors were relocated from one country to another country, thus with different vaccination schedules. Furthermore, a third dose of booster vaccine was introduced during the study period, again with different policies in different areas/countries. We thus included all donors with a documented vaccination history (i.e. type of vaccines, time of each dose, interval between the doses, if have infections before or after vaccinations) and grouped them accordingly. We have clarified the design of the study (Page 28, new supplementary Fig.1) and the reason for the separation of early and late phases based on 85 days (3 months) in replies to specific comments 4 and 5, respectively.

The biggest challenge is how to use these various data points to provide data to inform the scientific and public health communities. Showing post vaccine titers, without knowing pre-vaccine titers limits interpretation. I.e. how do we know that these groups are similar pre boost? There are some important questions in this field, but unfortunately the study as implemented is unable to contribute to a better understanding of heterologous boosts.

Reply: To address the pre-vaccine titer impact on the immune response after booster dose, we have added analysis of the longitudinal samples where we compared the pre- and post-vaccination titers in a group of individuals (Page 5, lines 114-115, and Page 31, Supplementary Fig4). We observed a similar increase in antibody response for individuals followed at different time points before vaccination and following each dose of mRNA vaccines (Supplementary Fig4A) as well as after the two doses of inactivated vaccines and the mRNA boost against VOCs (Supplementary Fig4B).

“Similar trends were recorded in an analysis of subset of individuals where longitudinal samples were available (Supplementary Fig.4).”

Also due to the concern of the reviewer as well as the small-sample size of this additional analysis, we have included this point in the limitations of the study (Page 11, lines 275-276).

“Furthermore, the data from various vaccination groups were mainly compared using cross-sectional analysis and longitudinal analysis was performed only in a subset of samples. Hence, our results should be confirmed in larger-scale longitudinal studies with different age groups and where the fluctuation of pre- and post-vaccination titers in specific individuals would be compared.”

Specific comments:

1. Abstract, line 41- increased 6 fold and 14 fold compared to what?

Reply: The compared groups have been clarified in the revised sentence (Page 2, lines 41-42).

“...compared to two doses of homologous inactivated vaccine”

2 March 2022

2. Abstract, line 42-44. The two sentences seem in contradiction. The heterologous tiers were the same as homologous, but heterologous significantly increased B cell response. Please clarify. Please note what you are measuring, i.e. pseudovirus neutralization titers, rather than using colloquial phrases like “level of specific antibodies.”

Reply: Based on this comment, we have clarified in the abstract that the phrase “significantly increased” is based on the comparison of booster third dose of heterologous vaccination compared to two-dose homologous inactivated vaccination. We have also clarified the immune components measured (Page 2, lines 47-48).

“...the RBD-specific memory B cell response and S1-specific T cell response compared to two or three doses of homologous inactivated vaccine”

3. Introduction can be considerably abbreviated. This section should set up the reader by reminding them of key facts, but is not generally meant as an exhaustive review. For examples, the review of all the available vaccines, % of world supply, and efficacy, doesn't help the reader understand heterologous boosts. Same with review of all the worlds variants.

Reply: We have shortened the introduction based on the suggestion of the reviewer (Pages 3-4).

4. Line 110 – why did you use 85 days to characterize early vs late boost.

Reply: We noted that antibody responses declined over time, and a significant reduction was observed after about three months post the second dose of the mRNA vaccines (Page 5, lines 106-108). Since a few volunteers followed longitudinally came for the three months follow-up sample just a few days before 90 days, we have set the cut-off at 85 days.

“We noted that antibody responses declined over time, and a significant reduction was observed about three months (85 days) post the second dose of the mRNA vaccines ($p \leq 0.0001$; Supplementary Fig.2 and Fig.3). Thus, all studied samples were further assigned to subgroups based on sampling time: the homologous vaccinated groups were further divided into two groups with an early or late sampling time, i.e., either less or more than 85 days after the second given dose”.

5. Line 220 – the design is not clear. There were 183 samples from 124 participants with vaccination schedule documented. What does that mean? The primary was documented? The boost was documented? or both?

Reply: We thank the reviewer for highlighting this point. The vaccination history documentation (type of vaccine, number of doses, interval between the doses, days after the latest dose at sampling, if have been infected) was performed in both primary and boost vaccination when the blood samples of an individual had been collected. We have now clarified this in the Method section.

Reviewer #2:

The submitted work describes heterologous vaccination strategies with the aim of improving response to the currently most active Omicron variant. This is quite important since this variant is exhibiting an immune escape phenotype and protection against infection is even reduced with the otherwise highly protective mRNA vaccines Comirnaty or SpikeVax. Since only a fraction of the global population has received those vaccines, strategies to increase the protection against Omicron are needed to improve world-wide vaccination supply.

The submitted manuscript describes such an approach, especially boosting of individuals that previously received two vaccinations with inactivated SARS-CoV-2 vaccines: BBIBP-CorV and CoronoVac. The administration of a single dose of Comirnaty did increase the vaccination response in those vaccinated individuals. Especially important is that a single vaccination of mRNA in those individuals reached specific antibody titer comparable to individuals that received three doses of mRNA vaccines. This indicates that

2 March 2022

such a third vaccination using mRNA might compensate for the lower immune response induced in those individuals previously receiving inactivated vaccine. However, at this stage it needs to be cautioned that three doses of the inactivated vaccines were not tested in this study and a comparison of vaccine potency of a such a third vaccination is not available.

I perceive this study as scientifically highly relevant for the ongoing SARS-CoV-2 pandemic and ways to improve global protection against SARS-CoV-2 ancestral or VOCs by heterologous vaccination.

Reply: We appreciate the reviewer's encouraging comments.

Moreover, for individual who got an inactivated vaccine, there is a need to know about the safety profile of a heterologous vaccination. As well this adds to the scientific knowledge. The data generation and interpretation is sound.

Reply: In the discussion, we have added a sentence and reference regarding the safety of the heterologous vaccination which is comparable to homologous vaccination based on the current data (Page 11, lines 265-267).

"The heterologous inactivated/mRNA prime-boosting strategy was also shown to be safe and cause fewer adverse reactions and events compared to other homologous and heterologous vaccination regimens⁵⁶." (Clemens, Lancet. 2022 Feb 5;399(10324):521-529).

Suggested Improvements:

Nevertheless, a weakness of this study is the absence of virus neutralization titers against ancestral and the most important variant Omicron, but for four data points displayed in supplemental figure 5. Please either generate more functional data in terms of VNTs or explain why this is not possible.

Reply: We have extended our determination of neutralization titers against G614 and VOC to cover all groups including the third dose of vaccination neutralizing antibodies against G614 and the Omicron variants (number of samples tested increased from 16 samples to 92 samples) (Page 23, Figure 2).

Moreover, please indicate how those samples were selected and to which extent they reflect the overall study population? Are this high/mid/or low responding individuals?

Reply: For neutralization titers, we have tried to include samples with longitudinal specimens available with different time points to make less bias toward comparison. Based on the suggestion of the reviewer, we have now provided a new supplementary Table S2 (Page 27) where we have shown that the samples included are not biased toward age, gender, as well as the titer of specific antibodies (Page 6, lines 143-145).

"Of note, the donors selected for this experiment were representative of the main study cohort with no significant differences in gender, age and specific antibody titers (Supplementary Table 2)."

The significance of RBD binding antibodies could be discussed more in details.

Reply: The importance of studying RBD as a target of the specific humoral immune response has been added to the discussion section based on this comment (Page 8, lines 188-192 and Page 9, lines 208-209).

Page 8, lines 188-192

"Binding of the RBD of the S protein to the ACE2 receptor is a critical initial step for SARS-CoV-2 to enter into target cells. Most neutralizing antibodies that are produced following infection or vaccination target the spike antigen on or proximal to the RBD and are considered important for blocking infection and viral clearance^{37,38}. Furthermore, studies have found that anti-RBD IgG antibody and neutralization titers correlate across all variants following infection or vaccination^{28,39}."

Page 9, lines 208-209

2 March 2022

“Mutations in the RBD may lead to reduction in neutralization susceptibility of VOCs by antibodies⁴¹. Therefore, anti-RBD antibody measurement can reflect the neutralization activity against emerging variants.”

In accordance with previously reported findings, we also found that the neutralizing ability of polyclonal plasma correlated positively with anti-RBD IgG titers for all 4 variants (see figure below for the reviewers)

Minor comments:

To which extent is the study population reflecting the age distribution in the countries where the inactivated vaccines are used? The study population seems rather to be young for a European population. However, relevant is the age distribution in which the third vaccination using an mRNA vaccine would potentially be applied. Please comment.

Reply: The sampling was based on volunteered blood donors who have been vaccinated by different vaccination strategies, which tend to be younger than the general population. The figure below compares the study population ages compared to the world population.

This has now been mentioned in limitations of the study (Page 11, line 268).

“The volunteers included in the study also tend to be younger than the general population.”

line 51-52: ...vaccination... is the main strategy to protect against... : not sure if really true since at least early in the pandemic social distancing or "lockdowns" have been used. Either indicate this is the main strategy since vaccines are available or make a disclaimer: "in the current stage of the pandemic"

Reply: We have added “In the current stage of the pandemic,...” and modified the text as “...one of the main strategies...” in the revised introduction (Page 3, line 52).

2 March 2022

line 86: comment on the mutations in spike RBD: the manuscript is describing Omicron as a variant with 15 mutations in spike, however, accurately variants with only 12 mutations (compared to ancestral spike) have been reported. So, already at this point Omicron is changing again.

Reply: We have added “according to the published original sequence” at the end of this sentence (Page 4, lines 87-88).

line 87: the BA.2 lineage of Omicron might even have a high transmissibility, since (even if slowly) increasing in areas where Omicron is predominant.

Reply: We have discussed the evolution of the virus and emergence of new lineage in the discussion, high transmissibility of BA.2 lineage of Omicron and its similar neutralization susceptibility by antibodies compared to BA.1 lineage has been discussed in the revised manuscript (Page 10, lines 249-256).

“As of February 2022, the Omicron variant, B.1.1.529 or BA.1, is constantly evolving and considered to have divided into four lineages: BA.1, BA.1.1, BA.2, and BA.3 with BA.1 accounting for most of the Omicron-cases worldwide but where BA.2 gaining ground and becoming the dominant form in some countries⁵⁰. BA.1, BA.2, and BA.3 share 12 RBD mutations, and BA.2 and BA.1 have 3 and 4 additional RBD mutations, respectively⁵¹. A recent study showed that neutralization of BA.2 by plasma from individuals vaccinated and boosted with the mRNA BNT162b2 reached titers only slightly (1.4-fold) lower than that against BA.1 suggesting that heterologous vaccination with inactivated and mRNA vaccines may still be efficient against new emerging variants⁵².”

line 118-119: "currently the most powerful immunization schedules to various variants including Omicron VOC". Please provide a reference for this statement.

Reply: Appropriate citations have been added (Page 5, lines 118, Refs 24-27).

line 127: Any ideas why the response would be stronger boosted by a third vaccination against Omicron, since this is the most divergent variant? Please comment.

Reply: “The potent anti-RBD antibody and neutralizing response against the Omicron variant observed following booster vaccination may be a consequence of affinity maturation following initial vaccination or SARS-CoV-2 infection with the ancestral S protein, increasing the affinity of existing neutralizing antibodies to counteract the effect of mutations in their target epitopes, and/or may be due to stimulation of low-level neutralizing antibodies that target conserved epitopes on the S protein^{26,43}. This may thus equalize the RBD specific antibody response and neutralization activity against the highly mutated Omicron variant and the original parent strain explaining the higher increase of RBD specific antibody response against Omicron (9.9-fold) compared to the original strain (6.3-fold) following boosting.” (Page 9, lines 214-221)

line 187: the neutralization data is available only for a very small subset of study participants. Please indicate this at this point of the manuscript.

Reply: We have now expanded our study on virus neutralization (from 16 samples to 92 samples) to cover all strategies of vaccination we have compared in this study (Page 23, Figure 2).

line 206: please explain in few words about why the real-world data from Chile is supportive in the context of this manuscript. Please comment.

Reply: Accordingly, complementary explanation regarding published real-world data from Chile has been integrated (Page 11, lines 263-265).

“...it is also supported by the recent real-world experience in Chile showing an increased effectiveness against COVID-19 and reduction of hospitalization using this strategy as compared to a homologous booster using the inactivated vaccine⁵⁵.”

line 213: please indicate why the lack of data for a third vaccination using inactivated vaccine is not a limitation of the study? Please comment.

2 March 2022

Reply: We thank the reviewer for indicating this important comment. We have now included one additional group of samples from donors who have received three-dose of homologous inactivated vaccine to the revised version of the manuscript (Page 5, lines 119-121, and Pages 22-24, Figures 1-3).

Reviewer #3:

In this work the authors have compared serum antibody ELISA titers against the receptor binding domain (RBD) from Wuhan-like SARS-CoV-2 with those against RBDs of VOCs, including omicron obtained from individuals vaccinated with inactivated COVID-19 vaccines, mRNA vaccines, and homologous or heterologous booster vaccination regimens (inactive + mRNA, mRNA + mRNA, convalescent + mRNA). Antibody ELISA titers against wild type and VOC RBD after mRNA booster was similar in individuals that were primed by inactivated virus vaccines, mRNA vaccines or in convalescent individuals. This correlated with enhanced levels of B cells and T cells as measured by ELISPOT.

Reply: We appreciate the reviewer's valuable comments.

Minor comment:

Line 77: The authors refer to Wuhan-like viruses that circulated early during the pandemic as "Wild type" virus. Ideally the authors should make clear in the beginning that wild type virus refers to Wuhan-like viruses that circulated before variants of concern started circulating.

Reply: Based on this comment, we have indicated which lineage has been used for the current study (Page 4, line 99). Although the RBD sequence is identical for both the Wuhan strains and G614 variant, we now refer the strain as G614 since the neutralization titers were determined against the G614 strain and we now included the antibody titers against the G614 spike (S1-S2 ectodomain).

Major comment:

In order to understand the relevance for protection of enhanced levels of antibody binding to RBD in ELISA assays, microneutralization assays with more sera (pseudo- or real viruses) need to be done. Also, antibodies that bind outside of the RBD can be protective. This should be measured as well, for example by ELISA against whole spike protein.

Reply: We have tested more sera in the neutralization assay using real viruses (Page 23, Figure 2). We have also conducted a new experiment to compare the antibody titers against the spike (S1-S2 ectodomain) of G614 and Omicron. (Page 6, line 135-138, Page 12, lines 305-307, and Pages 33-34, Supplementary Fig.6 and 7).

The authors have generated much data that is very interesting to the scientific community (for example the antibody decline over time and the microneutralization data, be it with limited groups size). The manuscript would benefit from it if these data were to be moved into the main text.

Reply: Thanks to the reviewer for this comment. The neutralization data have been moved to the main text (Page 23, Figure 2). The antibody decline over time has been left in the supplemental data since we consider that additional samples at later time points should be included to have a more accurate measure of the decline following heterologous vaccination (Page 30, Supplementary Fig3).

Reviewer comments, second round review:

Reviewer #2 (Remarks to the Author):

Thanks for revision of the original manuscript and addressing my and the comments from the reviewing colleagues.

Most valuable in my opinion is the addition of data for the three vaccinations of inactivated vaccines hence this allows for a fair evaluation of the heterologous (single) mRNA vaccination.

One additional comment, not sure if it is underscored sufficiently in the manuscript that with 1 mRNA vaccination as a boost (following two doses of a weaker inactivated vaccine) similar VNTs against ancestral as well as VOCs can be achieved as if three vaccinations using mRNA would be use. So, one (most properly more expensive) dose is sufficient to come to the "gold-standard" response and would be a good investment even in resource poor conditions to get the global population protected as good as possible against this still very harmful disease. I hope the global community will support the "most at need" populations to follow this path.

Minor comment:

line 275: ...general population... would suggest to change to ...average global population (or similar, indicating the statement "general" is neither referring to the populations in countries the study was conducted.

Reviewer #3 (Remarks to the Author):

My concerns have been addressed. This is substantially revised version of the manuscript.

10 April 2022

Reviewer #2 (Remarks to the Author):

Thanks for revision of the original manuscript and addressing my and the comments from the reviewing colleagues. Most valuable in my opinion is the addition of data for the three vaccinations of inactivated vaccines hence this allows for a fair evaluation of the heterologous (single) mRNA vaccination.

Reply: We appreciate the reviewer's comments.

One additional comment, not sure if it is underscored sufficiently in the manuscript that with 1 mRNA vaccination as a boost (following two doses of a weaker inactivated vaccine) similar VNTs against ancestral as well as VOCs can be achieved as if three vaccinations using mRNA would be use. So, one (most properly more expensive) dose is sufficient to come to the "gold-standard" response and would be a good investment even in resource poor conditions to get the global population protected as good as possible against this still very harmful disease. I hope the global community will support the "most at need" populations to follow this path.

Reply: In line with the comment, we have highlighted this important message in the conclusion section of the manuscript (Page 11, lines 247-251).

Minor comment: line 275: ...general population... would suggest to change to ...average global population (or similar, indicating the statement "general" is neither referring to the populations in countries the study was conducted.

Reply: We have corrected this phrase to "average global population" based on this precise comment (new Page 11, line 254).

Reviewer #3 (Remarks to the Author):

My concerns have been addressed. This is substantially revised version of the manuscript.

Reply: We appreciate the reviewer's comments.